# An Investigation into the Effects of Electric Field Uniformity on Electrospun TPU Fiber Nano-Scale Morphology

**DOI:** 10.3390/mi14010199

**Published:** 2023-01-13

**Authors:** Aaron Morehouse, Kelton C. Ireland, Gobinda C. Saha

**Affiliations:** Nanocomposites and Mechanics Laboratory (NCM Lab), University of New Brunswick, Fredericton, NB E3B 5A3, Canada

**Keywords:** electrospinning, parallel plate, single needle, electric field, beading, scanning electron microscopy

## Abstract

ANSYS Maxwell was used to replicate the conditions of two potential electrospinning configurations: a needle–plate and a parallel-plate configuration. Simulations showed that the electric field generated within the parallel-plate configuration was much more uniform than that within the needle–plate configuration. Both configurations were assembled and used electrospin fibers at three different spinning distances (10 cm, 12 cm, and 15 cm), at a consistent electric field strength of 1.7 kV/cm. Scanning electron microscopy was used to compare the morphologies of the fibers produced in both configurations in order to confirm whether a more uniform electric field yielded thinner fibers. The results show that the needle–plate configuration produced finer fibers than the parallel-plate configuration at all three spinning distances. However, there was no difference in the fiber diameters produced at the 12 and 15 cm spinning distances within the needle–plate configuration, implying thinning may only occur up to a certain distance in this configuration.

## 1. Introduction

Electrospinning is the process of drawing out fibers by exposing a conductive solution to a strong external electric field [1,2,3]. High voltages are applied to a spinneret through which the solution passes. As the voltage increases, the conductivity of the solution extrudes the fluid into an elongated droplet shape known as a Taylor cone. Once the charge concentration on the Taylor cone exceeds the surface tension of the fluid, a single jet is ejected, producing ultrafine fibers [2,3,4,5,6,7,8]. These fibers undergo further thinning as they propagate towards the grounded collection plate [2,3,6,8,9,10]. Fibers produced during electrospinning can reach diameters ranging from several nanometers to a few micrometers [1,11,12,13]. This makes them highly sought after due to their extensive applications in filtration and medical fields [1,5,6,8,9,11,12,13,14,15].

There are several factors which must be considered in order to achieve fiber formation. These factors can be divided into three categories of variables: ambient, solution, and processing variables. Ambient variables are those that are present in the space where electrospinning is being conducted (e.g., humidity and temperature) [2,8,10,13]. Solution variables, such as surface tension, conductivity, and viscosity, are specific to each solution composition. Finally, processing variables are those that are related to the physical electrospinning setup, such as voltage, spinning distance, and flow rate [2,4,5,6,7,8,10,13,15,16,17].

More complex electrospinning configurations will make use of multiple needles, for which an extensive understanding of the electric field morphology is required [18,19,20]. However, the scope of this study relates to ongoing research into the production of a thermoplastic polyurethane (TPU)-based face mask. A face mask produced from electrospun fiber mats would drastically improve the filtration capability of the mask, making it more effective against viruses. Current findings within this research show very sporadic values in Young’s modulus, which is believed to be due to non-uniform fiber morphology. In an effort to increase the strength and elasticity of the fiber mats, an investigation into the relationship between electric field uniformity and fiber morphology is proposed.

The electrospinning apparatus that was used in the following experiments is shown in Figure 1. The syringe pump is driven by a drive screw powered by a motor. As the screw turns, the syringe pump presses on the plunger of the syringe, ejecting solution. The flow rate of the solution is defined by the user. The solution passes through the spinneret, which has a high-voltage lead attached, which in turn charges the solution. The applied voltage is defined by the user, and ranges from 0 to 30 kV. The collection plate is grounded, which establishes an electric field originating from the needle to the collector. Since the electric field is being generated from a single point and extending to a plate, it is believed to be very non uniform. The first part of this study aimed to model the morphology of the electric field, and the second part aimed to examine the possible effects of a non-uniform electric field on fiber deposits.

Similar work has been conducted by Zheng et al. [14] with the modelling of electric fields around different spinneret configurations. In their research, they modelled the electric field around a single needle spinneret, as well as a disk-shaped spinneret with a diameter of 50 mm. Their results showed that the electric field around the needle spinneret was highly non uniform, with a very strong electric field close to the needle, which rapidly decayed when moving away from the needle. The electric field close to the disk-shaped spinneret was weaker but was much more consistent when moving away from the spinneret. Their results also showed that more uniform electric fields yielded shorter linear sections as well as a higher whipping frequency (more loops generated per second by electrical instability). With a shorter linear section, the fibers spun in the more uniform electric field were able to achieve smaller fiber diameters than those spun by the needle configuration.

Yang et al. [21] have also conducted similar work, investigating the effect of differences in electric field uniformity on electrospun fibers. In their study, they investigated the effect of electric field uniformity on several electrospinning parameters, such as the development of jet instability, jet length, whipping frequency, etc. Their results demonstrated that the fiber diameter was dependent on both solution properties and the uniformity of the electric field.

## 2. Experimental

The following study aims to expand upon content discussed by Yang et al. [21]. However, in their work, an aqueous solution of poly (ethelyne oxide) (PEO) was used as the polymer solution. The polymer of interest in the following study was TPU, which is not water-soluble; therefore, it will not be an aqueous solution like that of Yang et al. The exact chemical composition of the polymer solution will be discussed in detail later in this section.

The first goal of the study was to understand the electric fields produced within two different electrospinning configurations, namely a needle–plate configuration and a parallel-plate configuration. In order to understand the morphology of the electric field, a replica of each electrospinning apparatus was modeled using the finite element software, known as ANSYS Maxwell (ANSYS, Canonsburg, PA, USA). Figure 2 shows the physical and simulated needle–plate configuration, while Figure 3 shows the physical and simulated parallel-plate configuration. ANSYS Maxwell uses finite element modelling (FEM), which is performed by subdividing the model into smaller pieces, a process known as meshing. Maxwell’s equations are applied to each mesh component in order to compute the electric field vector at each point in space. These vectors are then combined to form a vector field, which describes the electric field within the prescribed region.

Measurements of the components of the electrospinning apparatus were taken in order to be modeled in ANSYS Maxwell. Table 1 summarizes the measurements taken. Fusion360 (Autodesk, San Rafael, CA, USA) was used to construct the 3D models. These models were then loaded into ANSYS Maxwell.

All components except for the needle were modeled using Fusion360. McMaster Carr (McMaster Carr, Elmhurst, IL, USA) was used to provide a CAD file for the needle due to its complex geometry. McMaster Carr states that the nozzle of the needle is constructed of stainless steel, and the base of the needle is made from polypropylene. The spinneret is also made from stainless steel, the collection plate is aluminum, and the high-voltage electrode is made from copper. Each component was assigned its constituent material in ANSYS in order to achieve accurate electric field calculations. ANSYS allows for the copper connector to be placed as an elevated voltage and for the collection plate to be grounded without the use of wires. It should be noted that the distance between the needle tip and the collection plate was set to 12 cm for both electrospinning configurations.

In order to run a simulation in ANSYS, the user must define a region surrounding the object(s) under investigation. For the simulations in this article, the defined region was specified as being filled with air. Since ANSYS builds the vector field by using FEM, a mesh must be defined. By default, ANSYS generates a mesh composed of 1000 elements. For this study, the number of elements was increased to 5000 to build a smoother mesh to produce a more continuous vector field. Figure 4 shows the dialogue box within ANSYS which allows the user to change the number of mesh elements.

For the second part of the study, the electrospinning setups simulated above were tested in order to study the fiber morphologies yielded in the different configurations. The electrospinning solution used was a 14 wt% thermoplastic polyurethane (TPU) solution dissolved in a mixture of dimethylformamide (DMF), ethyl acetate (EA), and lithium chloride (LiCl). The solution was provided with the electrospinning device from Inovenso (Inovenso, Istanbul, Turkey). The solution was used as received with no further refinement or modification. DMF is an extremely effective solvent for dissolving TPU and aids in increasing the conductivity of the solution [4,22]. A negative consequence of the use of DMF is that it is an extremely polar solvent, and therefore causes the solution to have a high surface tension [4,22,23,24]. High surface tension within the solution is a hindrance to the electrospinning process, since it can lead to jet breakup which leads to beading and non-uniform fibers [6,7,10,15,16,17]. The addition of EA aids in decreasing surface tension, but also decreases the conductivity of the solution [4]. It is believed that the addition of lithium chloride aids in increasing the solution conductivity to offset the negative side effects of the ethyl acetate.

The main objective of this study was to test the effect of electric field uniformity on final fiber morphology, and to do so, Equation (1) was used to compute electric field strength:(1)$$E=\frac{V}{d}$$
where *E* is the electric field strength, *V* is the applied voltage, and *d* is the distance between the charged piece and ground plate. Working with the parallel-plate configuration, it was found that jet formation began to appear at a distance of 10 cm with a flow rate of 0.25 mL/h and an applied voltage of 17 kV. This resulted in an electric field strength of 1.7 kV/cm. Electrospinning trials took place at 10 cm, 12 cm, and 15 cm. In order to retain consistency, the same electric field strength was used for the three different spinning distances. The same applied voltages and spinning distances were used for both electrospinning configurations. It should be noted that the above equation is typically used in an ideal scenario, where the electric field is perfectly uniform. It is believed that this is not the case for either configuration, and was therefore only used as a baseline to compare the different spinning configurations at a given distance. Table 2 summarizes the distances and the applied voltages to achieve an electric field strength of 1.7 kV/cm.

Three deposits were produced for each spinning distance and applied voltage for both the needle–plate configuration and parallel-plate configuration, with one exception. Only two samples at the 15 cm distance for the needle–plate configuration were produced due to complications arising from the electrospinner. Once the 17 samples were produced, they were analyzed under SEM (Scios 2 Field Emission) with the help of the UNB microscopy and microanalysis group. Three locations were captured on each electrospun sample stub with SEM at magnifications of 1000, 10,000, and 25,000 times. The 25,000-times magnification images were used to measure fiber diameter using the ImageJ software (National Institute of Health, Bethesda, MD, USA). Thirty measurements were taken on each of the 51 images at 25,000-times magnification. Analysis of these measurements was performed to statistically compare the difference in fiber morphology of the two different plate configurations for each given spinning distance.

## 3. Results

In order to replicate the conditions of the electrospinning setup as realistically as possible, the copper connector clamped between the spinneret in Figure 2 was assigned a voltage of 20 kV. The collection plate was grounded and placed 12 cm from the needle tip. The simulation was performed prior to the trialing with the parallel-plate setup; therefore, the exact electric field strength was not 1.7 kV/cm. Figure 5a–d show ANSYS Maxwell simulations of the electric field that was generated within the needle–plate configuration.

It is apparent from Figure 5a–d that the electric field that was generated in the needle–plate electrospinning configuration was highly non-uniform, which was expected. The non-uniformity was due to the electric field originating at a point and expanding towards a large plate. The electric field did reach a point of uniformity close to the collection plate; however, the strength of the field dropped significantly. Figure 6a–c below show the electric field generated within the parallel-plate configuration.

It is immediately apparent that the electric field was far more uniform both in direction and in magnitude. When comparing the strength of the fields from Figure 5 and Figure 6, there are areas in Figure 5 with stronger field regions (denoted by darker red vectors). These results agree with those found by Zheng et al. [14], who claimed that the electric field strength in the needle configuration is stronger around the needle, but rapidly decays when moving further towards the collection plate. The materials used for the collection plate, spinneret, and needle were unchanged from the previous configuration. The plate to which the spinneret is connected is made from aluminum, like the collection plate. In the proposed configuration, there was no need for a copper connector, since the entire plate to which the spinneret is connected was charged to 20 kV.

Of the 17 deposits produced, the first 9 were obtained using the parallel-plate configuration. All parallel-plate deposits were produced in a single day except for the first. The first deposit was produced the day before the others, where the average ambient humidity was 59.2%. On the second day, the ambient humidity ranged from a minimum of 77.9% to a maximum of 80.1%. Table 3 summarizes the average fiber diameters of all 17 deposits, and Figure 7 plots the average fiber diameters. The average fiber diameters and uncertainties were calculated from the three images taken for each deposit, each consisting of 30 measurements. Therefore, each deposit diameter is an average calculated from 90 measurements (in reality, most are 89 measurements due to the rejection of a single outlier in most cases).

Table 3 shows a significant difference in fiber diameter between deposit 1 and deposits 2 and 3. These deposits were produced with the same process parameters (17 kV, 10 cm, and 0.25 mL/hr). The only notable difference between these three deposits was the humidity at which they were produced. A two-sample z-score was calculated to statistically compare the fiber diameters, where it was found that the means for deposits 1 and 2 differed by 20.5 standard deviations and by 12.2 for deposits 1 and 3. For this reason, it was concluded that deposit 1 is statistically different from deposits 2 and 3. For comparison, the z-score found between deposits 2 and 3 is only 0.7. Figure 8, Figure 9 and Figure 10 show the fibers formed in deposits 1, 2, and 3, respectively.

The main scope of this study was to compare the fiber morphologies between the needle–plate and parallel-plate configurations. In order to do this, the average fiber diameter and associated uncertainties were found for each configuration at each spinning distance. These differences were then statistically compared. As stated previously, deposit 1 is statistically different to deposits 2 and 3 and is therefore not included in the calculation for the average fiber diameter at 10 cm for the parallel-plate configuration. Table 4 shows the average fiber diameter at each spinning distance for both electrospinning configurations. The z-score is the statistical comparison between the average fiber diameters at a given spinning distance for the two different spinning configurations. A z-score value exceeding two implies that the two data sets are in fact different, since their means are different by two standard deviations. This is equivalent to defining a confidence interval of 95%, or a significance level of α = 0.05, since 95% of a population’s data will fall within two standard deviations of the mean. The results from Table 4 show that the fibers produced in the needle–plate configuration are finer than those that were produced in the parallel-plate configuration.

Z-score values were also calculated between the adjacent spinning distances for a given electrospinning configuration to see if increasing spinning distance decreased fiber diameter. Values were computed between the 10 and 12 cm spinning distances and 12 and 15 cm spinning distances for both configurations. Table 5 summarizes the results, where PP denotes the parallel-plate configuration, and NP denotes the needle–plate configuration.

## 4. Discussion

By combining the results of Table 4 and Table 5, it is seen that the fibers that are spun in the parallel-plate configuration are thinner with increasing spinning distance. This might be explained by the fact that the fibers can undergo further whipping before hitting the collection plate. There is no statistical difference between the fiber diameters at the 12 and 15 cm distances in the needle–plate configuration. This may indicate that within the needle–plate configuration, the fibers solidify around the 12 cm distance and are then unable to thin any further as they continue to propagate towards the collector. Further testing would need to be conducted to support this theory, however.

The results from this study showed average fiber diameters that were considerably smaller than those found by Yang et al. [21]. Their findings reported that their tip-to-target configuration (which is similar to the needle–plate configuration in this study) yielded an average fiber diameter between roughly 500 and 700 nm. Similarly, they reported that their plate–plate configuration (similar to the parallel-plate configuration) yielded fiber diameters that ranged from 400 to 600 nm. There are several reasons why these results could vary. First of all, this study used a TPU/DMF solution for electrospinning, whereas Yang et al. used a PEO polymer dissolved in water. As discussed in the conclusions by Yang et al., the fiber diameter is not only dependent on the electric field uniformity, but also on the solution properties. Seeing as both the polymer and the solution are different in the experiments, the differences in solution parameters are most likely the cause of the differing fiber diameters. There are also major differences in the processing parameters used in both experiments. In this study, the electrospun jet was ejected horizontally, with spinning distances between 10 and 15 cm, while Yang et al. forced the jet to be ejected vertically downward at distances ranging between 35 and 40 cm. Finally, the applied voltages in this study ranged between 17 and 25.5 kV for both spinning setups, whereas the applied voltages by Yang et al. ranged between 7.5 and 15 kV for the tip-to-target configuration, and between 20 and 32.5 kV for the plate–plate configuration. By Equation (1), the electric field strengths were different between the two studies. Therefore, for the reasons listed above, a direct comparison between the fiber diameters produced in this study and that by Yang et al. cannot be made. There are simply too many parameters that are responsible for fiber diameter, and the parameters used in the two studies are too different.

Deposit 1 raises an interesting postulate that humidity might play an important role in the final fiber diameter that is achieved during electrospinning. In order to confirm this theory, further testing is suggested, with the ambient humidity around 60% in order to try and replicate the conditions of this deposition.

A study conducted by Raska et al. [25] investigated the effect of humidity on electrospun fibers. While their study focused on electrospinning silk fibroin (SF) and polyvinyl alcohol (PVA), the findings were similar to those in this study. In their study, they varied the humidity between 50 and 80% and found that increasing humidity yielded thicker fibers. This is what was observed between deposits 1 and 3; however, further testing must be performed in order to come to any definitive conclusions. The replication of conditions which yielded deposit 1 is seemingly important in order to potentially show a relationship between relative humidity and fiber diameter. Should the results from this follow-up study be in line with those found by Raska et al., it may suggest that relative humidity has a direct influence on average fiber diameter for electrospun fibers, regardless of their composition.

## 5. Conclusions

By using ANSYS Maxwell, it was confirmed that the electric field within the needle–plate configuration is highly non-uniform in comparison to that of the parallel-plate configuration. Since the electric field strength decayed rapidly in the needle–plate configuration when moving away from the needle, it was expected that the degree of whipping would decrease, resulting in larger fibers than for the parallel-plate configuration. Contrary to this belief, the results show that finer fibers were formed by the needle–plate configuration.

There are, however, a few different avenues for future work. In order to further investigate the finer fibers that were formed in deposit 1, another study should be designed to produce samples at the same parameters (17 kV, 10 cm, 0.25 mL/h, and ~60% humidity). Further testing could also be performed to try and confirm the theory that the fibers produced in the needle–plate configuration are only capable of thinning up to a certain distance.

## Figures and Tables

**Figure 1 micromachines-14-00199-f001:**
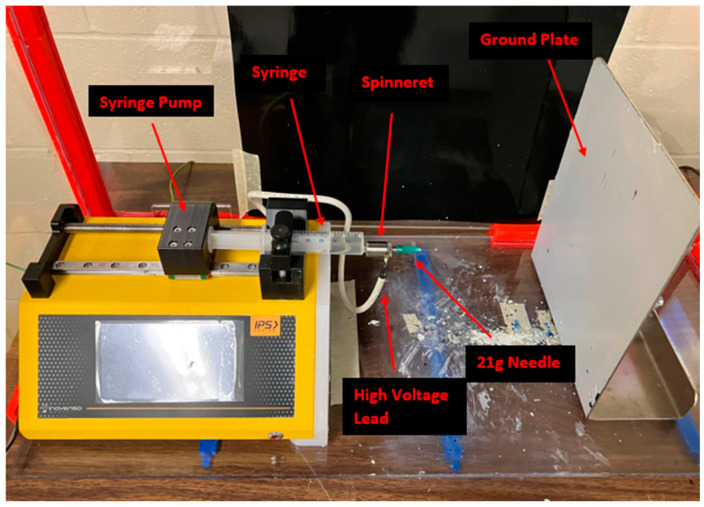
Electrospinning apparatus.

**Figure 2 micromachines-14-00199-f002:**
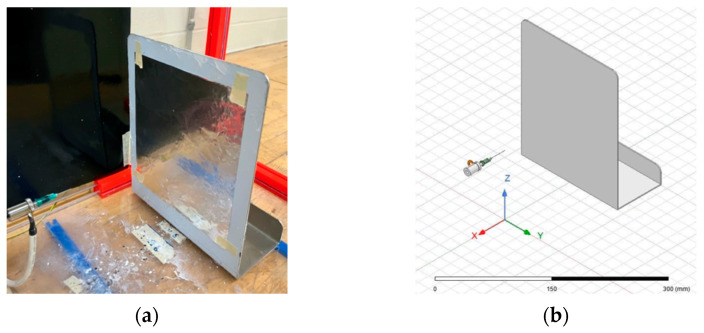
(**a**) Physical electrospinner needle–plate setup; (**b**) ANSYS simulation electrospinner needle–plate setup.

**Figure 3 micromachines-14-00199-f003:**
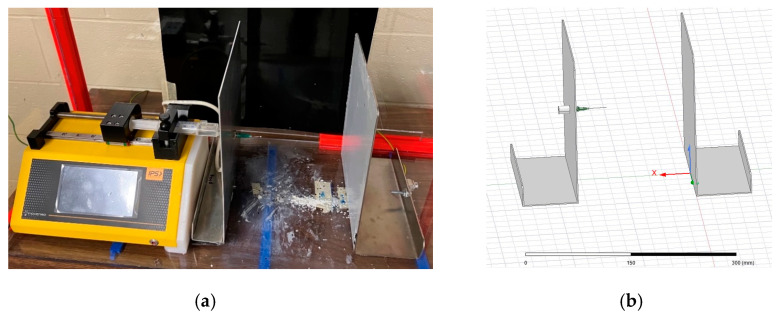
(**a**) Physical electrospinner parallel-plate setup; (**b**) ANSYS simulation electrospinner parallel-plate setup.

**Figure 4 micromachines-14-00199-f004:**
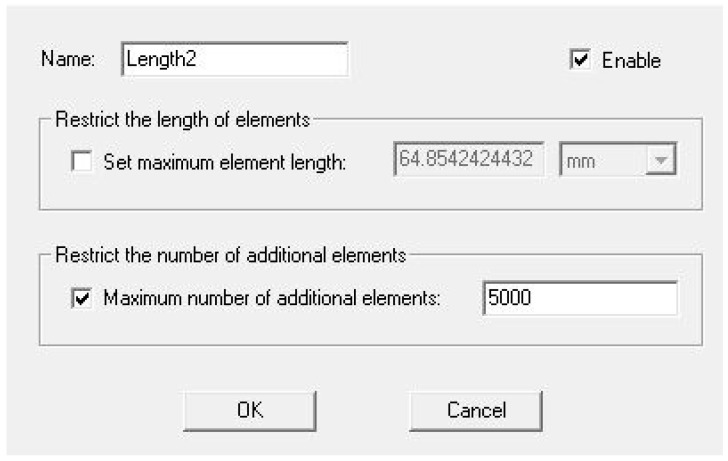
ANSYS Maxwell mesh generation dialogue box.

**Figure 5 micromachines-14-00199-f005:**
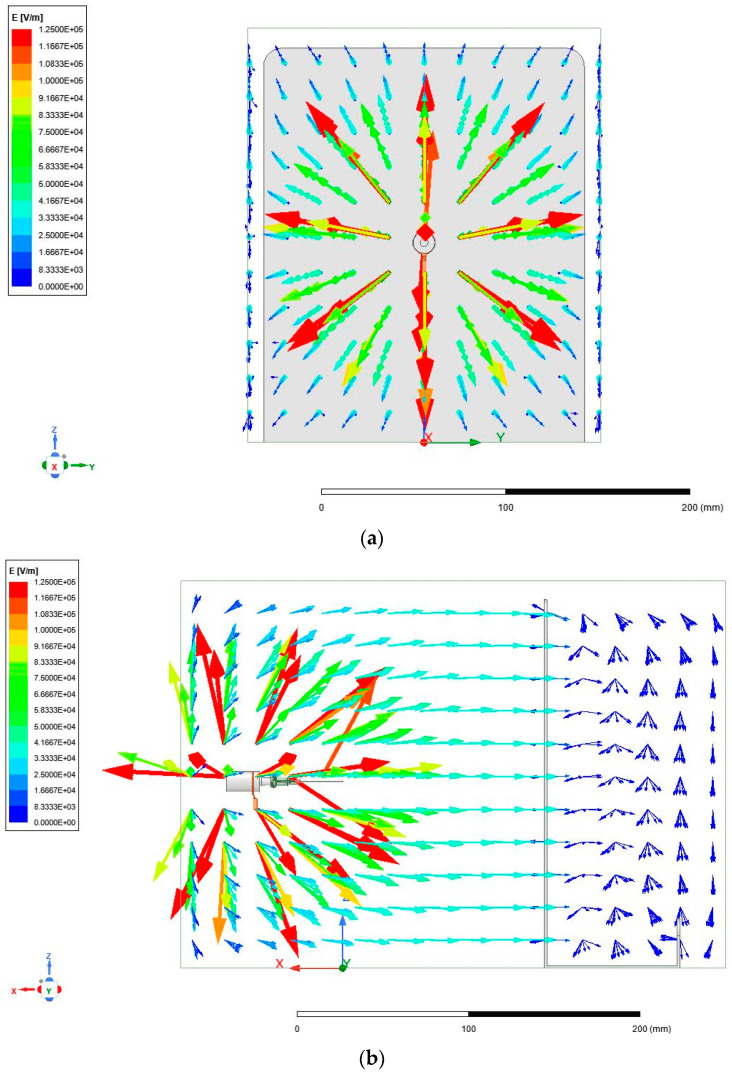
(**a**) Front view of electric field generated in needle–plate configuration. (**b**) Side view of electric field generated in needle–plate configuration. (**c**) Top-down view of electric field generated in needle–plate configuration. (**d**) Isometric view of electric field generated in needle–plate configuration.

**Figure 6 micromachines-14-00199-f006:**
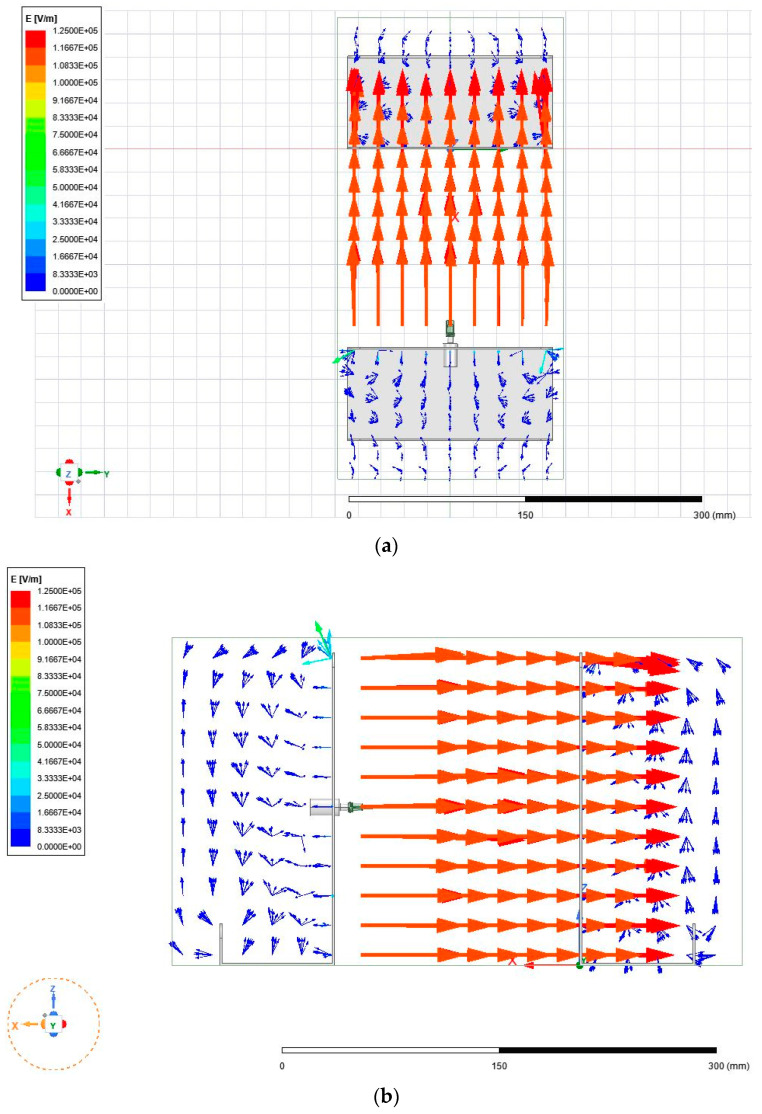
(**a**) Top-down view of electric field generated in parallel-plate configuration. (**b**) Side view of electric field generated in parallel-plate configuration. (**c**) Isometric view of electric field generated in parallel-plate configuration.

**Figure 7 micromachines-14-00199-f007:**
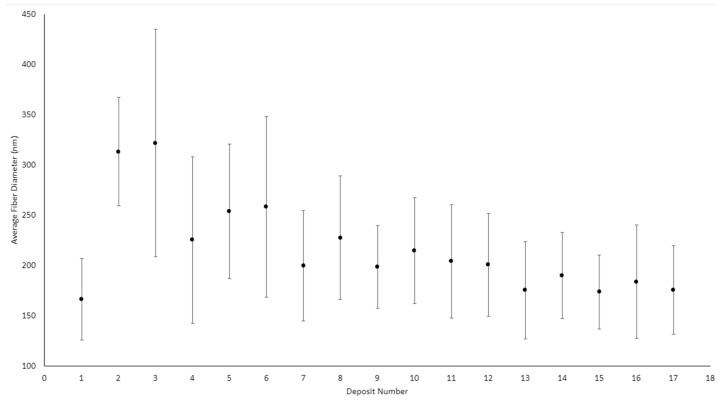
Average fiber diameters for parallel-plate configuration (deposits 1–9) and needle–plate configuration (deposits 10–17).

**Figure 8 micromachines-14-00199-f008:**
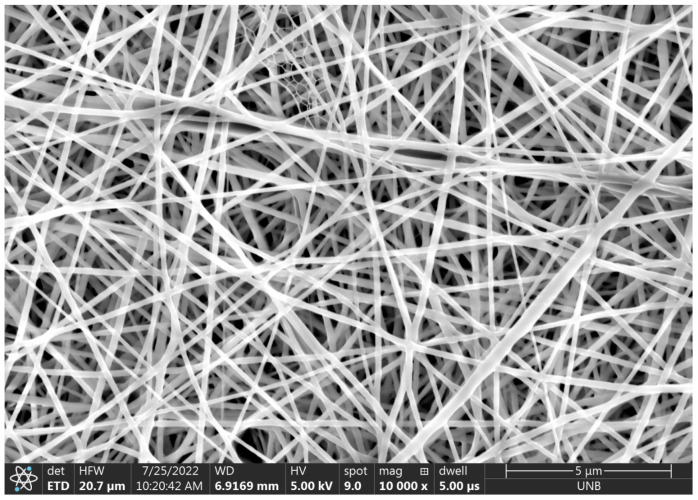
The 10,000× magnification of electrospun fibers from deposit 1.

**Figure 9 micromachines-14-00199-f009:**
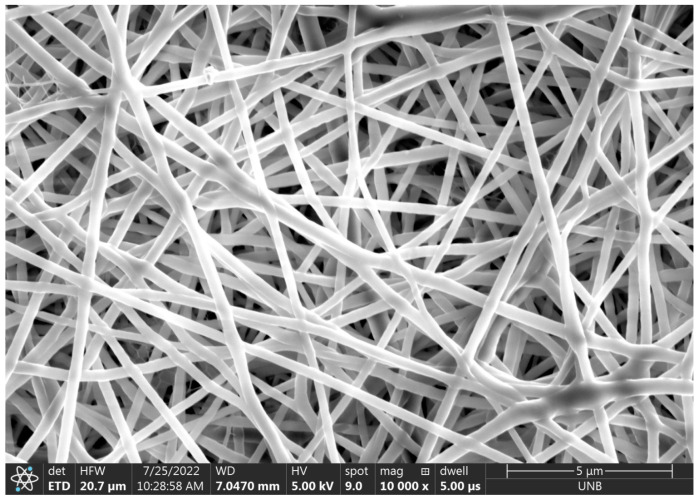
The 10,000× magnification of electrospun fibers from deposit 2.

**Figure 10 micromachines-14-00199-f010:**
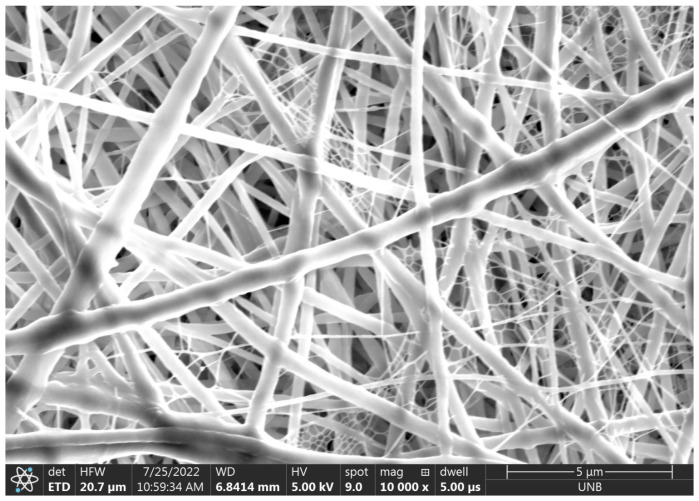
The 10,000× magnification of electrospun fibers from deposit 3.

**Table 1 micromachines-14-00199-t001:** Electrospinning setup measurements.

Component	Measurement
Collection plate width	7″
Collection plate height	8.625″
Collection plate thickness	0.0625″
Needle length	0.5″
Needle size	21 Gauge

**Table 2 micromachines-14-00199-t002:** Electrospinning distances and voltages to achieve 1.7 kV/cm.

Distance (cm)	Applied Voltage (kV)
10	17
12	20.4
15	25.5

**Table 3 micromachines-14-00199-t003:** Electrospun fiber diameters and associated uncertainties.

Deposit	Average Fiber Diameter (nm)	Uncertainty (nm)	Configuration	Distance (cm)
1	166	41	Parallel-plate	10
2	313	54	Parallel-plate	10
3	322	113	Parallel-plate	10
4	226	83	Parallel-plate	12
5	254	67	Parallel-plate	12
6	259	90	Parallel-plate	12
7	200	55	Parallel-plate	15
8	228	62	Parallel-plate	15
9	199	41	Parallel-plate	15
10	215	53	Needle–plate	10
11	205	56	Needle–plate	10
12	201	51	Needle–plate	10
13	176	48	Needle–plate	12
14	190	43	Needle–plate	12
15	174	37	Needle–plate	12
16	184	56	Needle–plate	15
17	176	44	Needle–plate	15

**Table 4 micromachines-14-00199-t004:** Average fiber diameters at each spinning distance for both spinning configurations.

	Parallel-Plate		Needle–Plate		Z-Score
Distance (cm)	Diameter (nm)	Uncertainty (nm)	Diameter (nm)	Uncertainty (nm)	
10	318	96	207	54	14.1
12	246	82	180	43	11.7
15	208	51	180	51	6.3

**Table 5 micromachines-14-00199-t005:** Comparison between average fiber diameters at adjacent spinning distances.

Comparison between Deposits	Z-Score
PP 10 and 12 cm	8.2
PP 12 and 15 cm	6.5
NP 10 and 12 cm	6.4
NP 12 and 15 cm	0.03

## Data Availability

The raw/processed data required to reproduce these findings cannot be shared at this time as the data also forms part of an ongoing study.

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
