# Peer review of "An Investigation into the Effects of Electric Field Uniformity on Electrospun TPU Fiber Nano-Scale Morphology"

_micromachines, 2023, doi:10.3390/mi14010199_

Round 1

Reviewer 1 Report

The manuscript reported the investigation of the effects of electric field uniformity on electrospun fiber Nano-scale morphology.

1. The title need be revised. “study” and “investigate” are duplicate.

2. The figures need be simplified. Some figures can be placed in a supporting information file.

3. Figure 5a, Figure 6b, Figure 7c, Figure 8d should be Figure 5a, Figure 5b, Figure 5c, Figure 5d? If this is true, then these four captions should be organized to a caption Figure 5. Figure 6a, Figure 6b, Figure 6c have similar problem. The authors can modified these points according to reported references.

4. Figure 8, Figure 9 and Figure 10 can be organized to a figure.

Author Response

Comments and Suggestions for Authors:

The manuscript reported the investigation of the effects of electric field uniformity on electrospun fiber Nano-scale morphology.

Comment #1: The title need be revised. “study” and “investigate” are duplicate.

Response: The author agrees that there was redundancy in the title, and a new title has been created: An investigation into the effects of electric field uniformity on electrospun TPU fiber Nano-scale morphology

Comment #2: The figures need be simplified. Some figures can be placed in a supporting information file.

Response: The Author disagrees. The figures should stay within the article for ease of access

Comment #3: Figure 5a, Figure 6b, Figure 7c, Figure 8d should be Figure 5a, Figure 5b, Figure 5c, Figure 5d? If this is true, then these four captions should be organized to a caption Figure 5. Figure 6a, Figure 6b, Figure 6c have similar problem. The authors can modified these points according to reported references.

Response: This issue appears to only exist in the PDF version, so there may be something happening during the conversion from word to PDF. Efforts have been made to fix this.

Comment #4: Figure 8, Figure 9 and Figure 10 can be organized to a figure.

Response: The author disagrees since these images are not of the same sample. If these were all images of one sample. It would be reasonable to label them as figures 8a-c.

Reviewer 2 Report

The article deals with the effect of the electric field uniformity, needle-plate distance and humidity on the morphology of fibers produced by electrospinning. 

I think the authors should face some issues before considering the work for publication. Most of all, I think they should give a much deeper introduction about the already existing scientific literature about the subject and compare their results to those published in literature.

Here are two critical points:

1- The effect of electric field of needle-plate and plate-plate configurations were already studied in detail in other papers (as an example, see

  • "Y.Tang et al., Effect of electric field distribution uniformity on electrospinning,

Journal of Applied Physics 103, 104307 (2008), https://doi.org/10.1063/1.2924439"), reporting results (fiber diameter values) in contrast with those of this paper. The authors should discuss the possible reasons for such different results and expand the literature coverage. 

2- The effect of humidity is presented as a novelty to be further studied in future works. However, there are already papers reporting similar effects (i.e.: A. Raksa, Piya-on Numpaisal and Y. Ruksakulpiwat Materials Today: Proceedings 47 (2021) 3458–3461). The authors should cite them in the discussion and explain how their findings represent a novelty to be published. 

Author Response

Comments and Suggestions for Authors:

The article deals with the effect of the electric field uniformity, needle-plate distance and humidity on the morphology of fibers produced by electrospinning.

I think the authors should face some issues before considering the work for publication. Most of all, I think they should give a much deeper introduction about the already existing scientific literature about the subject and compare their results to those published in literature.

Here are two critical points:

Comment #1: The effect of electric field of needle-plate and plate-plate configurations were already studied in detail in other papers (as an example, see

"Y.Tang et al., Effect of electric field distribution uniformity on electrospinning, Journal of Applied Physics 103, 104307 (2008), https://doi.org/10.1063/1.2924439"), reporting results (fiber diameter values) in contrast with those of this paper. The authors should discuss the possible reasons for such different results and expand the literature coverage.

Response: The author agrees that this should be mentioned. The author has read the suggested paper, and believes that the reason for differing results is most likely due to the different solution parameters. Tang et al. conducted their research using an aqueous PEO solution, while the research in question was conducted using a TPU polymer dissolved in DMF solution. Due to two different polymers being used, along with different solvents, it is not possible to make a direct comparison between fiber diameters. It should also be noted that other processing parameters were different, namely Tang et al. made use of a vertical jet along with a spinning distance between 35-40 cm, while the author used a horizontal jet and spinning distance between 10-15 cm. This is however a very important point, and has been discussed in detail throughout the research paper.

Comment #2: The effect of humidity is presented as a novelty to be further studied in future works. However, there are already papers reporting similar effects (i.e.: A. Raksa, Piya-on Numpaisal and Y. Ruksakulpiwat Materials Today: Proceedings 47 (2021) 3458–3461). The authors should cite them in the discussion and explain how their findings represent a novelty to be published.

Response: The author agrees. The paper in question has been read and is now included in the discussion. The author believes that the novelty of further studying the effects of humidity on fiber diameter on different polymer would help to show that humidity effects are consistent regardless of electric field uniformity or solution composition.

Reviewer 3 Report

In this work, ANSYS Maxwell was used to replicating the conditions of two potential electrospinning configurations: a needle-plate, and a parallel plate configuration. The simulation work was presented and discussed. There still existed some problems to be concerned about. My detailed comments are as follows:

  1. What’s the full name of ANSYS? Is it software?
  2. The authors should address the motivation of this work and the application background.
  3. Why should this study be detailed? What is the innovation in this work? The author proposed that similar work has been down in the last paragraph of the introduction part, what is the gap or shortcoming? How did your work intend to fill those gaps?
  4. Authors should also proofread their manuscript (some spelling and grammar errors). English grammar errors should be eliminated throughout the manuscript. A thorough check from a native English writer would be a good idea.
  5. Please double-check the name of the figures, Figure 5a? Figure 6b? Figure 7c?
  6. The resolution of the figures is not enough, and the letters are too small.
  7. A more and proper discussion about the results is necessary.
  8. Authors must add recent studies and compare the outcome of the perspective with much deeper elaborations. Also, authors can cite some recent publications.

Author Response

Comments and Suggestions for Authors:

In this work, ANSYS Maxwell was used to replicating the conditions of two potential electrospinning configurations: a needle-plate, and a parallel plate configuration. The simulation work was presented and discussed. There still existed some problems to be concerned about. My detailed comments are as follows:

Comment #1: What’s the full name of ANSYS? Is it software?

Response: ANSYS is simply a finite element modeling software. The author has made is more clear when introducing it that it is not an acronym, rather a software.

Comment #2: The authors should address the motivation of this work and the application background.

Response: The author has included motivation into the introduction

Comment #3: Why should this study be detailed? What is the innovation in this work? The author proposed that similar work has been down in the last paragraph of the introduction part, what is the gap or shortcoming? How did your work intend to fill those gaps?

Response: The work discussed in this article is a derivative of ongoing work within the research group, and should be further investigated in order to improve the material properties of the product being created. This has been further emphasized within the article.

Comment #4: Authors should also proofread their manuscript (some spelling and grammar errors). English grammar errors should be eliminated throughout the manuscript. A thorough check from a native English writer would be a good idea.

Response: A thorough reading has been conducted.

Comment #5: Please double-check the name of the figures, Figure 5a? Figure 6b? Figure 7c?

Response: There was an issue when converting from word document to PDF. It is believed to be corrected now.  

Comment #6: The resolution of the figures is not enough, and the letters are too small.

Response: The author agrees. Figures 5a-d have been increased in size to ease in reading, matching other figure sizes.

Comment #7: A more and proper discussion about the results is necessary.

Response: The Author agrees, a more in depth discussion has been added to discuss the findings.

Comment #8: Authors must add recent studies and compare the outcome of the perspective with much deeper elaborations. Also, authors can cite some recent publications.

Response: The author agrees, comparable studies have been include and comparisons have been made.

Reviewer 4 Report

Electrospinning has been well studied in the past decades, since Professor Reneker relaunched this research in 1990s. Please find the article below. 

Reneker, Darrell H., and Iksoo Chun. "Nanometre diameter fibres of polymer, produced by electrospinning." Nanotechnology 7, no. 3 (1996): 216.

Since then, a lot of experimental / simulation research had been performed and completed worldwide

Authors statement of " There has been extensive research into the morphology of the electric fields surrounding multiple needle electrospinner setups the, but very little in the way of single needle electrospinning electric field morphology", which is a false statement to the reviewer. Single needle spinning electric field had been studied even before a more complex multiple needle field, in 2000s.

Considering electrospinning has already been commercialized and used in industrial production, reviewer has concern of the originality and the contribution of this research to provide in scientifical and industrial field.

Author Response

Comments and Suggestions for Authors:

Electrospinning has been well studied in the past decades, since Professor Reneker relaunched this research in 1990s. Please find the article below.

Reneker, Darrell H., and Iksoo Chun. "Nanometre diameter fibres of polymer, produced by electrospinning." Nanotechnology 7, no. 3 (1996): 216.

Since then, a lot of experimental / simulation research had been performed and completed worldwide

Authors statement of " There has been extensive research into the morphology of the electric fields surrounding multiple needle electrospinner setups the, but very little in the way of single needle electrospinning electric field morphology", which is a false statement to the reviewer. Single needle spinning electric field had been studied even before a more complex multiple needle field, in 2000s.

Considering electrospinning has already been commercialized and used in industrial production, reviewer has concern of the originality and the contribution of this research to provide in scientifical and industrial field.

Response: The author agrees, the purpose of this specific research was not clearly listed. The motivation for this study is specific to ongoing research that is ongoing within the research group. The electrospun deposits will be used in face masks hopefully, but first the material properties must be understood. Currently, there is much deviation in Young’s modulus and ultimate tensile strength, therefore this study into the relationship between electric field uniformity and fiber morphology was proposed.

Round 2

Reviewer 2 Report

The work can be published now.

Author Response

Thank you, the authors appreciate your time in giving valuable feedback.

Reviewer 3 Report

The revised manuscript has been improved.

Author Response

The authors thank you for taking your time to provide feedback.

Reviewer 4 Report

Authors made some good revisions to help clarify the motivation of the on-going research. Appreciate that! Though, the originality of the research and the significance of scientific impact is still low to the review. Hate seeing research resources wasted in repeats works. Authors could do better in literature review to prevent this from future. 

In regarding face mask application, melt-blown material has been well developed for it vs. e-spinning has the major limitation of low production rate.  The review would be interested to see more clarification on research motivation.

Author Response

Comment # 1: In regarding face mask application, melt-blown material has been well developed for it vs. e-spinning has the major limitation of low production rate.  The review would be interested to see more clarification on research motivation.

Response: First of all, the authors are in full agreement with the reviewer that the melt-blown materials have been well-studied and are being used in many applications. However, as a filter material they are non-effective due to their pore size. Typically, the pores are on the micron-scale, which is large enough for viruses to pass through. This was the motivation for the research work presented in this paper. The paper reports the creation of smooth fibers through the use of a specific thermoplastic urethane (TPU) polymer in a mixture of Dimethylformamide/Ethyl acetate (DMF/EA) using the electrospinning method. We believe that the use of the specific polymer is a novel component in the reported paper.

As part of an ongoing research project, the presented data would provide the basis for technology transfer wherein upscale manufacturing would be explored.